# UM-164, a Dual Inhibitor of c-Src and p38 MAPK, Suppresses Proliferation of Glioma by Reducing YAP Activity

**DOI:** 10.3390/cancers14215343

**Published:** 2022-10-29

**Authors:** Huizhe Xu, Ye Zhang, Jia Liu, Jing Cui, Yu Gan, Zhisheng Wu, Youwei Chang, Rui Sui, Yi Chen, Ji Shi, Haiyang Liang, Qiang Liu, Shulan Sun, Haozhe Piao

**Affiliations:** 1Central Laboratory, Cancer Hospital of China Medical University, Cancer Hospital of Dalian University of Technology, Liaoning Cancer Hospital & Institute, No.44 Xiaoheyan Road, Dadong District, Shenyang 110042, China; 2Department of Neurosurgery, Cancer Hospital of China Medical University, Cancer Hospital of Dalian University of Technology, Liaoning Cancer Hospital & Institute, No.44 Xiaoheyan Road, Dadong District, Shenyang 110042, China; 3Institute of Cancer Stem Cell, Dalian Medical University, No.9 Lvshun South Road, Lvshunkou District, Dalian 116044, China

**Keywords:** UM-164, glioma, YAP, proliferation, c-Src, p38

## Abstract

**Simple Summary:**

UM-164, as a high-potency c-Src inhibitor, is the original lead compound to be developed for targeting triple-negative breast cancer. Here we validated the fact that UM-164 induces the inhibition of glioma cell proliferation, migration and spheroid formation, as well as cell cycle arrest in the G1 phase. Moreover, UM-164 triggers YAP translocation to the cytoplasm, reduces the activity of YAP, and decreases the expression levels of CYR61 and AXL, which are generally known as YAP target genes. We further demonstrated that p38 appears to play a greater role than Src in the declined YAP activity mediated by UM-164. Additionally, UM-164 restrains glioma growth in vivo. Therefore, our data provide the first evidence that UM-164 exerts anti-tumor outcomes in glioma via the Hippo-YAP signaling pathway.

**Abstract:**

UM-164 is a dual inhibitor of c-Src and p38 MAPK, and has been a lead compound for targeting triple-negative breast cancer. UM-164 shows stronger binding to the active sites of Src compared with the conventional Src inhibitor Dasatinib. While Dasatinib has displayed some inhibitory effects on glioma growth in clinical trials, whether UM-164 can suppress glioma growth has not been reported. Here we show that UM-164 suppressed the proliferation, migration and spheroid formation of glioma cells, and induced cell cycle arrest in the G1 phase. Moreover, UM-164 triggered YAP translocation to the cytoplasm and reduced the activity of YAP, as evidenced by a luciferase assay. Accordingly, UM-164 markedly decreased the expression levels of YAP target genes CYR61 and AXL. Importantly, ectopic expression of wild-type YAP or YAP-5SA (YAP constitutively active mutant) could rescue the anti-proliferative effect induced by UM-164. Intriguingly, p38 MAPK appears to play a greater role than Src in UM-164-mediated inhibition of YAP activity. Furthermore, the in vitro anti-glioma effect mediated by UM-164 was confirmed in a xenograft glioma model. Together, these findings reveal a mechanism by which UM-164 suppresses the malignant phenotypes of glioma cells and might provide a rationale for UM-164-based anti-glioma clinical trials.

## 1. Introduction

Glioma, derived from glial cells, is the most common primary tumor of the central nervous system [1]. The benefit of standard treatment for patients with glioma (surgery combined with radiotherapy and chemotherapy) has reached a bottleneck due to its high malignancy. At present, the commonly used molecular targeted drugs for glioma include anti-tumor alkylating agents, small molecule protein kinase inhibitors and humanized monoclonal antibodies, such as temozolomide, imatinib and bevacizumab [2,3,4]. However, prolonged use of these drugs can lead to problems such as reduction of white blood cells and platelets, as well as proteinuria [5,6,7]. Therefore, it is urgent to search for novel therapeutic drugs.

In recent years, studies have shown that glioma development can be inhibited by targeting various signaling pathways, including PI3K/AKT/mTOR, Hippo, MEK/ERK and p38 MAPK [8,9,10,11,12]. As a crucial protein in the development of multiple pathways, Src is a homologous gene of the sarcoma virus oncogene v-Src in cells, and is the first proto-oncogene encoding a non-receptor tyrosine kinase to be discovered [13]. With further research, high levels and protein activity of c-Src have been observed in cancer. It is known that c-Src plays a key role in multiple signaling pathways of cell proliferation, migration and angiogenesis [14]. Src is also closely related to the occurrence and development of glioma [15,16,17,18]. Accordingly, interfering with the expression or activity of Src could inhibit the proliferation, migration, invasion and growth of glioma [19], highlighting that Src could be a therapeutic target for glioma. Nonetheless, conventional Src inhibitors including Dasatinib and Bosutinib have not demonstrated significant efficacy in phase II clinical trials for glioma stem cells (GSCs) or recurrent GBM due to tumor heterogeneity and the differential influence of Src family kinases on glioma growth [20,21,22]. Combining Src inhibitors with compounds targeting other key regulators or signaling pathways, such as BI2536 (PLK1 inhibitor), KX2-361 (Tubulin inhibitor) and JSI-124 (STAT3 inhibitor), has achieved notable consequences for glioma treatment in vitro and in vivo [23,24,25,26]. UM-164 (also known as DAS-DFGO-II) is a highly potent c-Src inhibitor that can bind to the inactive kinase conformation of c-Src and is more potent than Dasatinib in its Src active site binding ability [27]. Because UM-164 can alter c-Src localization in triple negative breast cancer (TNBC) cells and displays anti-TNBC cell proliferative activity, UM-164 is a promising lead compound for the treatment of TNBC [28]. In addition, UM-164 can also effectively inhibit the activation of p38α and p38β MAPK. However, the anti-tumor activity of UM-164 in glioma has not been reported.

YAP (Yes-associated protein) is a core effector of the Hippo pathway that is activated by translocation from the cytoplasm to the nucleus, thereby regulating gene expression and promoting tumorigenesis. A transcriptional activation domain (PPXY) at the carboxyl terminus of YAP indicates that YAP has an auxiliary transcriptional activity as a transcription coactivator [29,30]. The four amino acid residues at the hydroxyl end of YAP can bind to PDZ-domain-containing proteins, such as ZO-2, and this interaction causes YAP to accumulate in the nucleus. Therefore, the binding of YAP to the PDZ domain is considered to regulate the nucleoplasmic transport of YAP in cells [31,32]. In the nucleus, YAP cannot bind DNA directly and it exerts its transcriptional activity mainly by binding to TEADs, a family of proteins containing the TEA domain, for transcription of AXL, CYR61, CTGF and other target genes.

The upstream kinases of the Hippo pathway, such as LATS1/2, negatively regulate the transcriptional co-activation ability of YAP by retaining it in the cytoplasm and inducing its degradation [33]. MST1/2 phosphorylation activates LATS1/2, which subsequently phosphorylates YAP or its homolog TAZ. The phosphorylation of YAP/TAZ by LATS1/2 occurs at serine/threonine residues in the HxRxxS motif. S127 (S89 corresponding to TAZ) and S381 (S311 corresponding to TAZ) play an important role in YAP localization. Meanwhile, the YAP protein is relatively stable, and its protein level is regulated mainly depending on its localization [34].

Src and YAP are coexpressed in normal organs and tissues, and the interaction between the two is crucial for maintaining the physiological function of normal cells [35]. Importantly, upon Src activation, YAP is activated as a result of LATS1 activity repression by the directly phosphorylation of LATS1 at tyrosine residues [36,37]. Moreover, several previous studies have shown that the Src inhibitor Dasatinib could inhibit the expression of YAP in a variety of cancer cells, including renal cell carcinoma, breast cancer and non-small cell lung cancer. In addition, JNK and p38 MAPK signaling cascades regulate YAP nuclear expression in PNX-induced alveolar regeneration, and JNK or p38 MAPK inhibitors could block the induction of YAP nuclear expression [38]. Therefore, as a dual inhibitor of Src and p38, the activity and cellular localization of YAP by UM-164 attracted our attention.

In this study, we investigated the impact of UM-164 on glioma cell growth. We show for the first time that UM-164 hijacks the Hippo/YAP pathway to exert anti-glioma activity.

## 2. Materials and Methods

### 2.1. Cells and Plasmids

HMC, LN229 and SF539 cell lines were obtained from the American Type Culture Collection (ATCC), cultured in Dulbecco’s modified Eagle’s medium (DMEM) supplemented with 5% or 10% fetal bovine serum (FBS) according to the instructions. The patient-derived glioblastoma cells GBM1492 were a gift from Dr. Zhiyou Fang, Hefei Institutes of Physical Science, Chinese Academy of Sciences. GBM1492 spheroids were cultured in a three-dimensional (3D) culture system supplied with DMEM/F12 medium supplemented with 2% B27, 20 ng/mL EGF and 20 ng/mL bFGF. All cells were incubated at 37 °C in a 5% CO_2_ incubator. Plasmids (PQCXIH-vector, PQCXIH-YAP, PQCXIH-YAP-5SA, V5-vector, V5-Src-wt, V5-Src-Y527F, Flag-vector and Flag-p38) were kind gifts from Dr. Songshu Meng, Dalian Medical University.

### 2.2. Antibodies and Reagents

Antibodies against Cyclin D1, CDK2, CDC6, pSrc, Src, pp38, p38, pYAP, YAP and AXL were purchased from Cell Signal Technology. Antibodies against CYR61 were purchased from Santa Cruz Biotechnology. Antibodies against Ki67 were purchased from Bioworld Technology. Antibodies against Lamin A+C, GAPDH and α-Tubulin, as well as secondary antibodies, were purchased from Huabio Biotechnology. UM-164 was purchased from Selleck. Cell Counting Kit-8 (CCK8) was purchased from Apexbio Technology. Nuclear and cytoplasmic extraction reagents were purchased from ThermoFisher Scientific.

### 2.3. Cell Viability, Colony Formation, Cell Migration and 3D Culture Assays

Cells were respectively seeded at a density of 2000 cells/well into 96-well cell culture plates and then treated with UM-164 at different concentrations for the indicated time. Cell viability was measured with CCK8 according to the manual. Cells were seeded into 6-well plates and treated with UM-164 for the colony formation assay. For the cell migration assay, cells were resuspended in 200 μL serum-free media and seeded into the upper chamber after pre-treatment with mitomycin C and UM-164; the complete medium with a total volume of 700 μL was added into the lower chamber. Then, cells were fixed with 4% paraformaldehyde and stained with 0.2% crystal violet. The 3D culture assay was performed as previously described [25].

### 2.4. Flow Cytometric Analysis

Cells were treated with vehicle or UM-164 for 24 h, and then harvested and fixed with ice-cold 70% ethanol overnight at 4 °C (LN229 cells), or 3.7% paraformaldehyde for 15 min (SF539 cells). Cells were incubated with PBS containing 50 µg/mL propidium iodide (PI), 100 µg/mL RNase A and 0.02% (*w*/*v*) TritonX-100 for 30 min at 37 °C in the dark. All samples were analyzed by flow cytometry.

### 2.5. Nuclear and Cytoplasmic Extraction

Cells were digested and harvested by centrifugation at a speed of 500× *g* for 5 min. Nuclear and cytoplasmic extraction was performed with the kit purchased from Thermo according to the instructions. Antibodies against Lamin A/C and tubulin were used as references for nuclear cytoplasmic proteins, respectively.

### 2.6. Immunoblotting, Immunofluorescence and Immunohistochemistry

Immunoblotting, immunofluorescence and immunohistochemistry were performed as previously described [39]. Details of the proportion of antibodies used are provided in Appendix A.

### 2.7. Quantitative Real Time PCR (qRT-PCR)

Total RNA was extracted from cells pretreated with vehicle or UM-164. cDNA was obtained by using the PrimeScript™ RT reagent Kit (Takara, Kusatsu, Japan), and qRT-PCR analysis was performed using TB Green^®^ Premix Ex Taq™ (Takara) and the CFX96TM Optics Module following the manufacturer’s instructions. The relative transcription levels of the genes were calculated using the delta-delta-Ct (ΔΔCT) method (expressed as 2^−ΔΔCT^) and normalized to GAPDH as an endogenous control. Primers are shown as follows:
axl: F-AACCTTCAACTCCTGCCTTCTCG R-CAGCTTCTCCTTCAGCTCTTCACcyr61: F-CTTACGCTGGATGTTTGAGTGTR-AGACTGGATCATCATGACGTTCTctgf: F-GCTTACCGACTGGAAGACACGR-CGGATGCACTTTTTGCCCTTcyclin d1: F-AGCTCCTGTGCTGCGAAGTGGAAACR-AGTGTTCAATGAAATCGTGCGGGGTgapdh: F-CTTCACCACCATGGAGGAGGCR-GGCATGGACTGTGGTCATGAG

### 2.8. RNA-seq

LN229 cells were treated with 100 nM UM-164 or DMSO for 24 h, and then the total RNA of the cells was extracted using Trizol. RNA-seq was performed by LC Bio Technology Co., Ltd. (Hangzhou, China). The total RNA quantity and purity were analyzed using a Bioanalyzer 2100 and RNA 6000 Nano LabChip Kit (Agilent, Santa Clara, CA, USA, 5067-1511); high-quality RNA samples with RIN number > 7.0 were used to construct the sequencing library. Gene differential expression analysis was performed with DESeq2 software between two different groups (and with edgeR between two samples). The genes with a false discovery rate (FDR) parameter below 0.05 and an absolute fold change ≥ 2 were considered differentially expressed genes. The RNA-seq dataset was deposited in the Gene Expression Omnibus (GEO) with accession number GSE213586.

### 2.9. Xenograft Tumor Models

Six-week-old male nude mice (Beijing Vital River Laboratory Animal Technology Co., Ltd., Beijing, China) were subcutaneously inoculated in the flank with LN229 cells (3 × 10^6^ cells in 100 μL PBS/mouse) to induce tumor development. When tumors reached an average volume of 100 mm^3^, mice were randomly divided into 3 groups and intraperitoneally (i.p.) injected with saline, 5 mg/mL or 10 mg/mL UM-164 every three days. Mice were sacrificed after 6 weeks, and tumor sessions were fixed for immunohistochemistry analysis or lysed with lysis buffer for WB assays. All the animal experiments have been approved by the Animal Ethics Committee of Dalian Medical University.

### 2.10. Statistical Analysis

Differences between experimental groups were evaluated by one-way ANOVA or the Student test, using the Graphpad Prism 8 software package to analyze the results. Statistical significance was based on a *p*-value of 0.05.

## 3. Results

### 3.1. UM-164 Inhibits Glioma Cell Growth

We first evaluated the effect of UM-164 on glioma cell growth with the CCK8 assay. The results showed that UM-164 treatment significantly inhibited cell growth of two glioma cell lines, LN229 and SF539, compared with vehicle treatment, with a more potent effect observed in SF539. The half-maximal inhibitory concentration (IC50) values of UM-164 at 24, 48 and 72 h were 10.07, 6.20 and 3.81 µM, respectively, in LN229 cells, and 3.75, 2.68, and 1.23 µM, respectively, in SF539 cells (Figure 1A). Therefore, 50 and 100 nM concentrations of UM-164 were used to treat glioma cells in subsequent cell experiments.

We also investigated the anti-tumor effects of UM-164 on GBM1492 patient-derived glioblastoma cells. As illustrated in Figure 1B, UM-164 markedly inhibited the growth of human primary glioma cells with an IC50 of 10 µM at 24 h. However, UM-164 treatment did not significantly affect the growth of human microglial cells (Figure 1B).

In Figure 1C, the cell colony formation experiments revealed that UM-164 could effectively inhibit the proliferation of glioma cells in a dose-dependent manner. In addition, transwell migration assays showed that the migratory abilities of LN229 and SF539 cells were greatly reduced after treatment with 100 nM UM-164 for 24 h (Figure 1D). To examine the effect of UM-164 on 3D glioma spheroids, we plated glioma cells in ultra-low attachment cell culture plates incubated with a 3D culture medium for 12 days. The results showed that in the presence of UM-164, the numbers and sizes of the LN-229- and SF539-derived spheroids were significantly reduced compared with vehicle treatment (Figure 1E).

### 3.2. UM-164 Induces Glioma G1 Phase Cell Cycle Arrest

To investigate the mechanism by which UM-164 inhibits glioma cell proliferation, we determined whether the inhibitory effect of UM-164 on the tested glioma cells is related to cell cycle progression by flow cytometry analysis. As shown in Figure 2A, treatment of LN229 cells with 50 or 100 nM UM-164 significantly increased the number of cells in G1 phase from 64.95% (control) to 71.44% and 75.79%, respectively. The number of cells in the G1 phase increased from 51.64% to 60.80% and 66.68%, respectively, in UM-164-treated SF539 cells, indicating that UM-164 induces cell cycle arrest.

We further conducted a Western blot assay to determine the effect of UM-164 on the protein levels of cell cycle regulatory proteins such as cyclin D1, CDK6 and CDC6. Figure 2B showed that these G1/S phase regulators are consistently and significantly decreased in both LN229 and SF539 cells following incubation with increasing doses of UM-164 (Figure 2C). In addition, a time-course expression pattern of these cell-cycle regulators was also observed (Figure 2D,E). We found that the protein levels of cyclin D1, CDK6 and CDC6 were reduced in UM-164-treated glioma cells in both dose- and time-dependent manners.

### 3.3. UM-164 Restrains YAP Nuclear Localization and Its Downstream Signaling

Since UM-164 is a highly effective Src and p38 MAPK inhibitor, we examined the changes in the phosphorylation levels of p38 and Src in glioma cells in response to UM-164, and the Src inhibitor Dasatinib was included as a control [40,41]. As shown in Figure 3A, p-p38 and p-Src (Y416) protein levels were simultaneously significantly decreased after UM-164 treatment of glioma cells. However, Dasatinib treatment resulted in a marked increase in p38 MAPK activation in LN229 and SF539 cells, although Src phosphorylation levels were more significantly decreased in these cells.

To further explore the mechanism of action of UM-164 in glioma cells, we performed RNA sequence (RNA-seq) analysis of LN229 cells treated with 100 nM UM-164 or vehicle. This analysis showed that UM-164 treatment significantly upregulated a total of 172 genes and downregulated 171 genes (|log2FC| ≥ 1 & q < 0.05) in LN229 cells (Appendix A). Of note, a number of Hippo pathway-related and YAP target genes were among those downregulated (Figure 3B). Next, quantitative real-time PCR was performed to examine the transcriptional change in four potential YAP target genes (cyr61, ctgf, ccnd1 and axl) in glioma cell lines upon UM-164 treatment. Indeed, UM-164 significantly reduced the mRNA levels of cyr61, ctgf, ccnd1 and axl in both LN229 and SF539 cells (Figure 3C). In Figure 3D, exposure to UM-164 in glioma cells led to elevated YAP phosphorylation at serine 127 or 397, accompanied with a decrease in the protein levels of YAP targets CYR61 and AXL, indicating an inhibitory effect of UM-164 on YAP activation and activity.

Furthermore, immunofluorescence (IF) assays revealed that YAP nuclear localization (green) was significantly reduced upon exposure to UM-164 treatment for 24 h (Figure 3E), while YAP retention in the cytoplasm was substantially increased compared with the vehicle control (arrow). Next, we examined YAP protein levels in UM-164-treated glioma cells using a nuclear and cytoplasmic extraction kit. In line with the IF results, the level of YAP protein in the nuclei was significantly reduced with UM-164 treatment (Figure 3F). We wondered if Dasatinib exhibits the same effect on YAP activity as UM-164. Dasatinib increased the phosphorylation level of YAP at serine 127, but the effect was much lower than that of UM-164. Likewise, UM-164 was more effective than Dasatinib in reducing the expression of the YAP downstream proteins CYR61 and AXL (Figure 4A and Appendix A). These results demonstrate that Dasatinib is not as potent as UM-164 for attenuating YAP target proteins.

### 3.4. UM-164 Hinders Glioma Cell Proliferation via the Hippo-YAP Pathway

To prove a critical role of YAP in UM-164-induced inhibition of glioma cell growth, we transfected PQ-Vector, PQ-YAP-WT and PQ-YAP-5SA into two glioma cells and then exposed these cells to 100 nM UM-164 for 24 h. YAP-WT or YAP-5SA, which is a constitutively active mutant, could provoke the expression of YAP downstream genes. Consequently, both YAP-WT and YAP-5SA could prevent the reduction of AXL and CYR61 protein levels induced by UM-164 (Figure 4B and Appendix A). Moreover, ectopic expression of YAP-WT or YAP-5SA also significantly restored the colony-forming ability of the cells compared to the vector control (Figure 4C and Appendix A).

In the canonical Hippo pathway, YAP is phosphorylated by the upstream MST1/2-LATS1-2 kinase cascade. We pretreated glioma cells with XMU-MP-1, an MST1/2 inhibitor [42,43]. As shown in Figure 4D,E and Appendix A, XMU-MP-1 significantly reversed UM-164-mediated inhibition of YAP target gene expression as well as inhibition of growth in LN229 and SF539 cells, indicating that the Hippo-YAP pathway is involved in the inhibitory effect of UM-164 on glioma cell proliferation.

### 3.5. p38 MAPK Is More Prominent in Blocking UM-164-Mediated Anti-YAP Activity

Given that UM-164 is a dual inhibitor of c-Src and p38, we next asked whether UM-164 inhibits glioma cell growth mainly via c-Src and/or p38. To clarify this, we transfected V5-Vector, V5-Src-WT and V5-Src-Y527F or Flag-Vector and Flag-p38 WT into LN229 and SF539 cells, and then treated the cells with UM-164. The results in Figure 5A showed that V5-tagged Src and its constitutively activate mutant partly rescued the UM-164-induced reduction of AXL and CYR61 abundance in glioma cells. Noteworthy, ectopic expression of p38 MAPK completely blocked the downregulation of YAP target proteins by UM-164 (Figure 5B), indicating that p38 MAPK is responsible for the UM-164-induced decrease in the levels of AXL and CYR61. Accordingly, p38 MAPK overexpression significantly reverted UM-164-repressed colony sphere formation in LN229 and SF539 cells compared to vector control (Figure 5C,D). Together, these results suggest that p38 MAPK but not Src might be the key mediator in UM-164-mediated anti-YAP activity and suppression of glioma cell growth.

### 3.6. UM-164 Suppresses Glioma Xenograft Growth

To substantiate our in vitro findings, a nude mouse xenograft model was established by subcutaneous injection of LN229 cells. Fifteen tumor-bearing mice were randomly divided into three groups to be administered saline and UM-164 at a dose of 5 mg/kg or 10 mg/kg by intraperitoneal injection every three days. As depicted in Figure 6A,B, the mice treated with UM-164 showed an obvious trend of persistent atrophy compared with the control group, and the effect of the 10 mg/kg group was superior to that of the 5 mg/mL group.

Ki67 staining confirmed decreased proliferation in tumors treated with UM-164 (Figure 6C,D). Likewise, the expression of cyclin D1 and CYR61 in tumor tissues was markedly reduced, as examined by immunohistochemistry analysis and Western blotting, respectively (Figure 6C–F). All these results confirmed that UM-164 had a strong inhibitory effect on glioma cell growth in the nude mouse xenograft model.

## 4. Discussion

Glioma is an aggressive cancer that can develop in the brain or spinal cord. Previous studies have demonstrated that changes in various signaling molecular pathways play an important role in occurrence and development of glioma. Based on preclinical studies, it is known that activation of Src and/or p38MAPK signaling pathways promotes glioma development and progression [44,45,46]. By utilizing potent inhibitors to target Src or p38MAPK-related signaling pathways, the proliferation and migration of glioma cells could be greatly inhibited [47]; therefore, the development of compounds targeting both Src and p38MAPK simultaneously is promising in anti-glioma therapy.

The Hippo pathway is an evolutionarily conserved signaling pathway that regulates organ size development, regeneration and carcinogenesis in multicellular organisms. Its abnormality is closely related to the occurrence and development of tumors. YAP and TAZ, the two coactivators involved in the Hippo pathway, are found to be highly active in a variety of cancers due to their roles in regulating cell proliferation and apoptosis, and when dysfunctional, lead to malignant tumors [48]. Recent studies have shown that the Hippo signaling pathway is regulated by upstream regulators, such as the p38MAPK and ERK/JNK signaling pathways [49,50]. Salloum et al. reported that YAP activation in nonalcoholic fatty liver disease (NAFLD) may be driven by FFA-induced p38/MAPK activation [51]. In addition, YAP contains an SH3-binding domain that can directly bind to the non-receptor tyrosine kinase Src [52]. Studies also showed that Src activation mediates YAP dysregulation, invasiveness and drug resistance in tumors [53,54]. UM-164 was discovered as a potent Src inhibitor. Surprisingly, it was found to inhibit p38MAPK activity. As a novel dual inhibitor of Src and p38MAPK, research on UM-164 as a tumor therapeutic agent is still missing and its mechanism of action in tumors remains to be explored.

Here, our study shows for the first time that UM-164 can reduce the proliferation of glioma cells by inhibiting the activity of YAP. We further demonstrate that UM-164 induces the translocation of YAP protein from the nucleus and thus inhibits YAP activation and decreases the expression of YAP downstream target genes. In addition, we also verified the theme of this study, namely that inhibition of the Hippo-YAP pathway is the main mechanism of the effects of UM-164 on glioma cells, through diverse in vitro experiments. In addition to the direct inhibitory effect of UM-164 on Src and p38MAPK, this study likewise indicated that UM-164 increases the phosphorylation of YAP. Compared with Dasatinib, which is also a Src inhibitor, UM-164 has shown that it could inhibit the oncogenic activity of YAP and its target protein more effectively.

Our results indeed demonstrated that compared with Src, p38 played a more prominent role in UM-164-treated glioma cells. However, considering the influence of many factors, including the transfection population, cells’ ability to tolerate expression of certain proteins and many more, we are not trying to emphasize the uniqueness of p38 MAPK. Furthermore, we believe that Src, p38 MAPK and even other underlying mechanisms that we have not found in this paper might play important roles in UM-164-induced inhibition of glioma cells and YAP oncogenic activity.

Of note, our in vitro findings have been firmly substantiated in the in vivo mouse model experiments. In our in vivo tumor model, a low dose (5 mg/kg) of UM-164 showed a reliable inhibitory effect on glioma. The use of a low dose can also minimize the toxicity of the drug to animals. Tissue immunohistochemistry and protein detection results have repeatedly and effectively verified that UM-164 could limit the proliferation of glioma cells and arrest the glioma cell cycle in vivo. Consequently, the dominant anti-glioma activity of UM-164 in vivo highlights its potential value in clinical applications.

## 5. Conclusions

Together, our data reveal a mechanism by which UM-164 suppresses glioma cell growth and underscores UM-164 as a promising agent for the treatment of glioma patients.

## Figures and Tables

**Figure 1 cancers-14-05343-f001:**
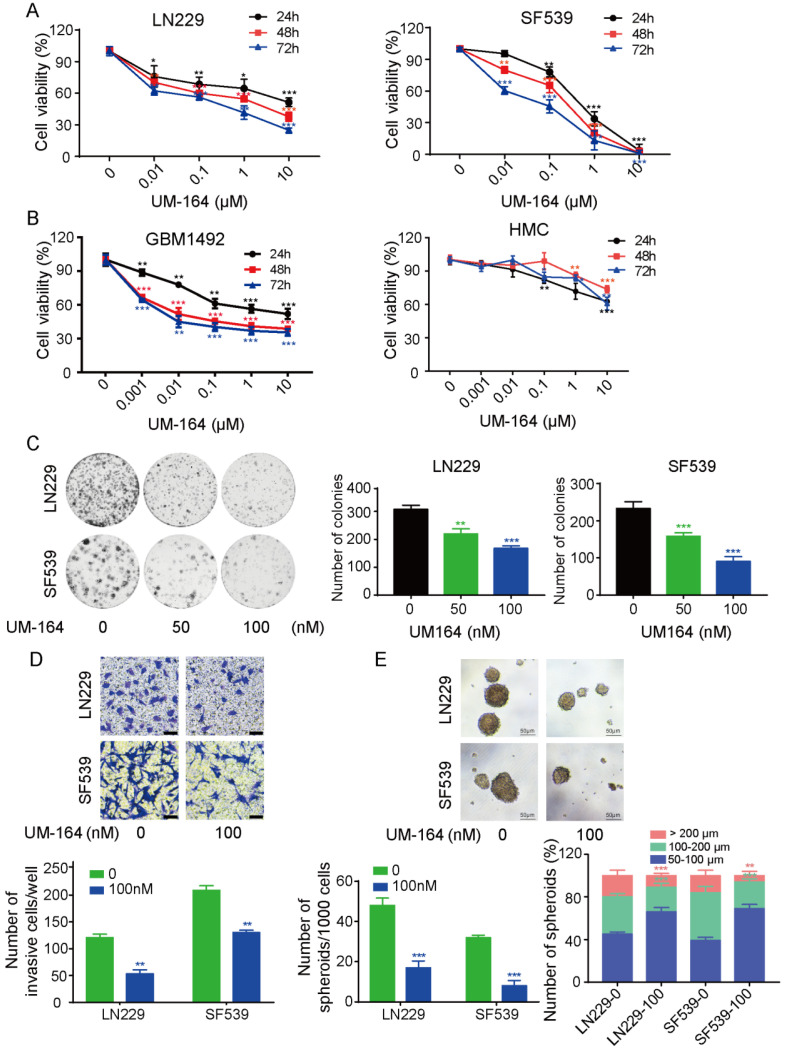
UM-164 inhibits glioma cell viability, proliferation, migration and spheroid formation capability. (**A**) Glioma cell lines LN229 and SF539 were treated with UM-164 at the indicated concentrations (0, 0.01, 0.1, 1, 10 μM) for 24, 48 and 72 h. Cell viability was measured by the CCK8 assay. (**B**) Primary glioma cells (GBM1492) and human microglial cells (HMCs) were treated with UM-164 at the indicated concentrations (0, 0.001, 0.01, 0.1, 1, 10 μM) for 24, 48 and 72 h. Cell viability was measured by the CCK8 assay. (**C**) LN229 and SF539 cells were treated with vehicle control or 50 nM and 100 nM UM-164 and cultured in complete medium for 14 days. The colony formation assay was performed. The number of colonies was counted and presented as the mean ± standard deviation. (**D**) LN229 and SF539 cells were incubated with vehicle control or 100 nM UM-164 and cell migration was examined by the transwell assay. The cells migrating to the bottom of the upper chamber were fixed and stained. The number of migrating cells was counted and measured. (**E**) Spheres of the LN229 and SF539 glioma cell lines treated with vehicle or 100 nM UM-164 in ultra-low-attachment plates. The number and size of spheroids were counted and measured, respectively. All experiments in this figure were performed three times with comparable results. * *p*< 0.05, ** *p* < 0.01, *** *p* < 0.001.

**Figure 2 cancers-14-05343-f002:**
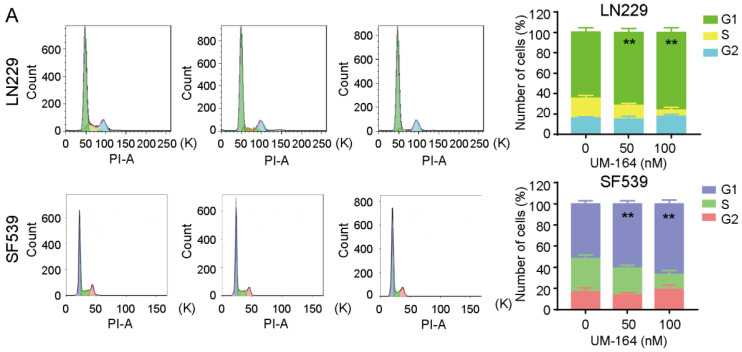
UM-164 regulates the glioma cell cycle. (**A**) LN229 and SF539 cells were treated with vehicle or 100 nM UM-164 and were stained with propidium iodide for cell cycle analysis by flow cytometry at 24 h (left). The percentages of the cell cycle distribution were presented as mean ± SEM (right). (**B**,**C**) LN229 and SF539 cells were treated with vehicle control, 50 nM and 100 nM UM-164 for 24 h; protein levels of Cyclin D1, CDK2 and CDC6 were detected by an immunoblotting (IB) assay. The ratios of Cyclin D1, CDK2 and CDC6 expression relative to GAPDH were represented. (**D**,**E**) LN229 and SF539 cells were treated with vehicle control and 50 nM UM-164 for 24 h or 48 h; protein levels of Cyclin D1, CDK2 and CDC6 were detected by an IB assay. The ratios of Cyclin D1, CDK2 and CDC6 expression relative to GAPDH were represented. Data are represented as mean ± SEM from three independent experiments, * *p* < 0.05, ** *p* < 0.01, *** *p* < 0.001. The uncropped bolts are shown in Appendix A.

**Figure 3 cancers-14-05343-f003:**
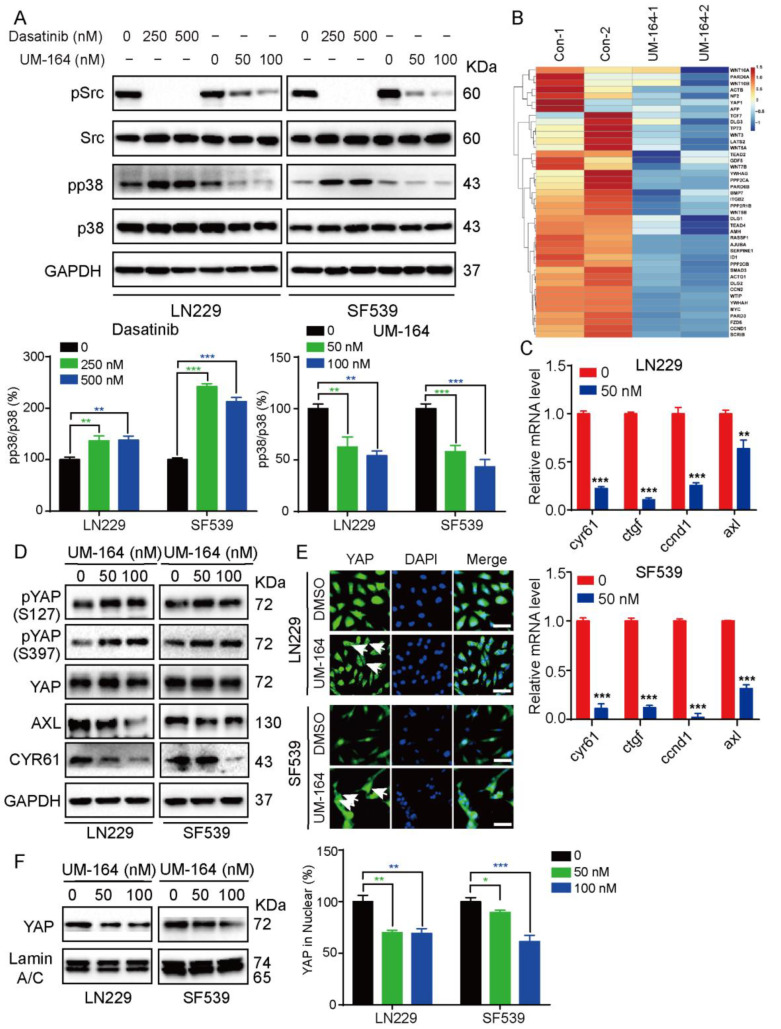
UM-164 downregulates YAP activity. (**A**) LN229 and SF539 cells were treated with Dasatinib (0, 250 and 500 nM) or UM-164 (0, 50 and 100 nM) for 24 h; protein levels of Src and p38 and their phosphorylation species were detected by an IB assay (top). p38 phosphorylation relative to total p38 was quantified (bottom). (**B**) The genes related to the Hippo-YAP pathway that were differentially expressed in LN229 cells exposed to vehicle control or 100 nM UM-164 are shown. (**C**) Real-time quantitative PCR analysis of the mRNA levels of four YAP downstream genes (cyr61, ctgf, cyclind1 and axl) in LN229 and SF539 cells with vehicle control and 50 nM UM-164 treatment. (**D**) IB analysis for YAP, p-YAP, CYR61 and AXL in LN229 and SF539 cells treated with vehicle and 50 nM or 100 nM UM-164. (**E**) Immunofluorescence (IF) staining for YAP (green) in LN229 and SF539 cells after UM-164 treatment for 24 h. Nuclei were stained with DAPI (blue). The white arrows represent the cellular location of YAP. Representative merged images are also shown for fluorescence signals. Scale bar = 25 μm. (**F**) Nucleoprotein in LN229 and SF539 cells treated with 50 or 100 nM UM-164 was isolated with a nuclear and cytoplasmic extraction kit, and protein levels of YAP in the nucleus were detected by IB analysis (left). The percentages of YAP in the nucleus relative to Lamin A/C were represented (right). Mean ± S.D. for three independent experiments is shown. * *p* < 0.05, ** *p* < 0.01, *** *p* < 0.001. The uncropped bolts are shown in Appendix A.

**Figure 4 cancers-14-05343-f004:**
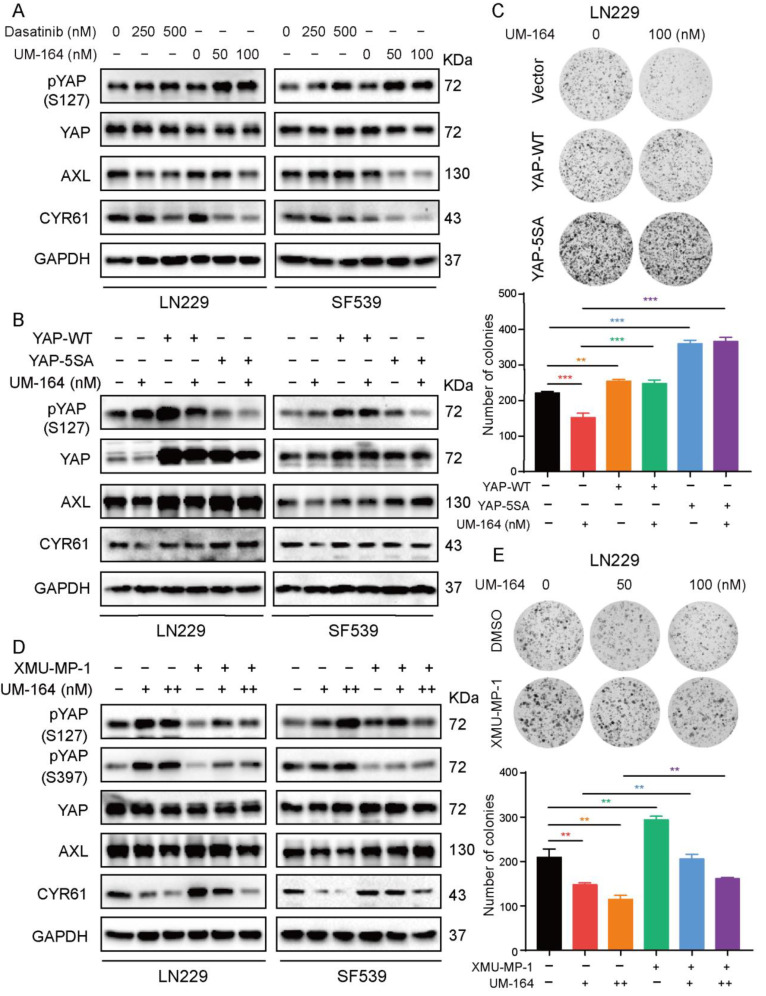
UM-164 suppresses glioma cell growth via the Hippo-YAP pathway. (**A**) LN229 and SF539 cells were incubated with Dasatinib and UM-164 at the indicated concentration for 24 h; protein levels of YAP, pYAP (S127), AXL and CYR61 were detected by IB. (**B**) LN229 and SF539 cells transfected with PQ-vector, YAP-WT or YAP-5SA were treated with vehicle or 100 nM UM-164. Protein expression of YAP, pYAP (S127), AXL and CYR61 was detected by IB. (**C**) Colony formation assays in LN229 cells were performed after the UM-164-exposed cells were infected with YAP-WT and YAP-5SA. The number of colonies was measured. (**D**) LN229 and SF539 cells were pretreated with 1 μM XMU-MP-1 and then treated with vehicle or 100 nM UM-164 for 24 h. Protein expression for YAP, pYAP(S127), pYAP(S397), AXL and CYR61 was detected by IB. (**E**) Colony formation assays in LN229 cells were performed after incubation with XMU-MP-1 and UM-164. The number of colonies was measured. All experiments in this figure were performed three times with comparable results. Data are represented as mean ± S.D. ** *p* < 0.01, *** *p* < 0.001. The uncropped bolts are shown in Appendix A.

**Figure 5 cancers-14-05343-f005:**
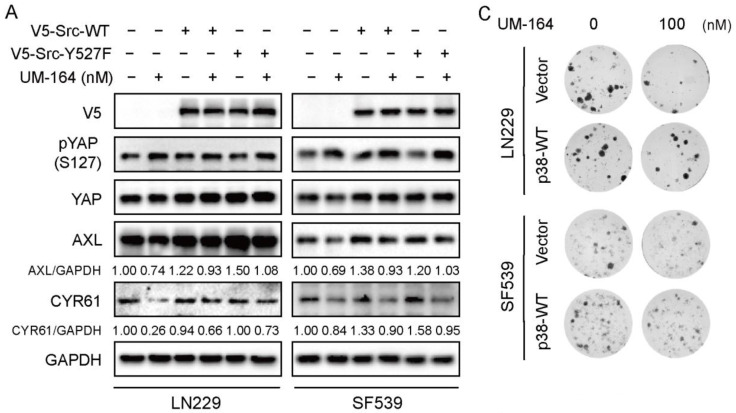
Inactivation of p38 plays a crucial role in UM-164-mediated inhibition of YAP activity. (**A**) LN229 and SF539 cells were transfected with V5-Vector, V5-Src-WT and V5-Src-Y527F, and then treated with vehicle or 100 nM UM-164 for 24 h. IB analysis of pYAP, YAP, AXL and CYR61 expression. The ratios of AXL and CYR61 expression relative to GAPDH were represented. (**B**) LN229 and SF539 cells were transfected with Flag-Vector and Flag-p38, and then treated with vehicle or 100 nM UM-164 for 24 h. IB analysis of pYAP, YAP, AXL and CYR61 expression. The ratios of AXL and CYR61 expression relative to GAPDH were represented. (**C**,**D**) LN229 and SF539 cells were transfected with Flag-Vector and Flag-p38, and then treated with vehicle or 100 nM UM-164; analysis of the ability to form colonies. All experiments in this figure were performed three times with comparable results. Data are represented as mean ± S.D. * *p* < 0.05, ** *p* < 0.01, *** * p* < 0.001. The uncropped bolts are shown in Appendix A.

**Figure 6 cancers-14-05343-f006:**
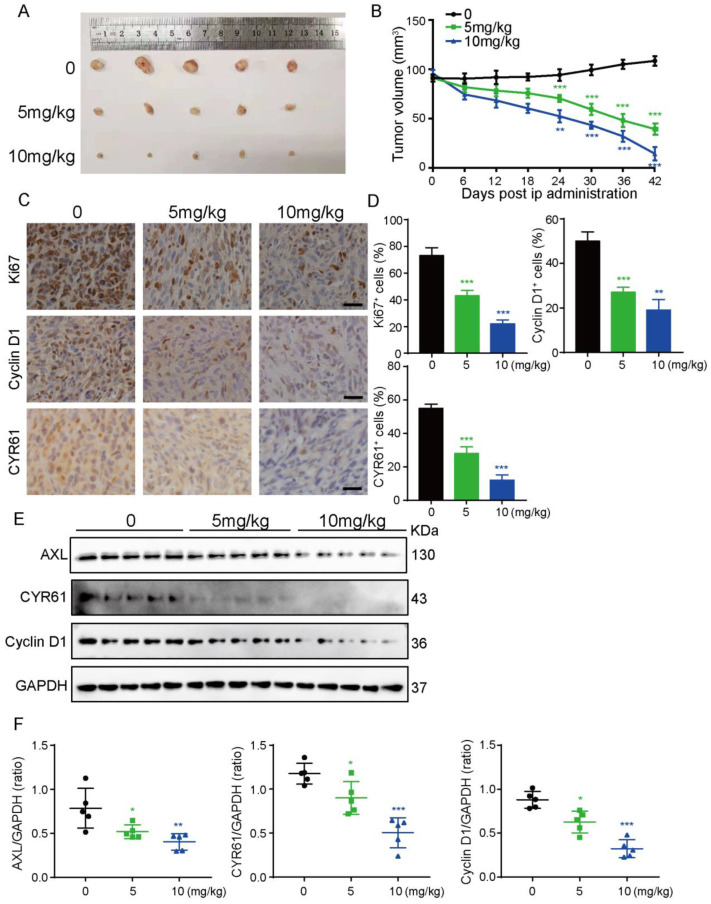
UM-164 shows anti-glioma proliferation capability in vivo. (**A**,**B**) Images and volumes of mice tumors at day 42 derived from subcutaneous injection of 3 × 10^6^ LN229 cells treated with UM-164 (5 mg/kg, 10 mg/kg) or saline administered by intraperitoneal injection (5 mice per group). The graph shows the change in tumor volume with respect to the initial treatment at day 0. (**C**,**D**) Immunochemistry (IHC) analyses of Ki67, Cyclin D1 and CYR61 expression were performed. Scale bar = 50 μm. The ratios of Ki67-, Cyclin D1- and CYR61-staining cells in the LN229 glioma xenografts were counted. (**E**,**F**) Protein levels of AXL, CYR61 and Cyclin D1 in glioma tumor xenografts were detected by IB. The ratios of AXL, CYR61 and Cyclin D1 relative to GAPDH were shown. Data are represented as mean ± S.D. * *p* < 0.05, ** *p* < 0.01, *** *p* < 0.001. The uncropped bolts are shown in Appendix A.

## Data Availability

All data reported in this paper will be shared by the lead contact upon request.

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
