# Peer review of "UM-164, a Dual Inhibitor of c-Src and p38 MAPK, Suppresses Proliferation of Glioma by Reducing YAP Activity"

_cancers, 2022, doi:10.3390/cancers14215343_

Round 1

Reviewer 1 Report

In this study, the authors show that UM-164 can suppresses proliferation of glioma  via Hippo-YAP signaling pathway. This is a potentially interesting study. However, there are several major points needed to be addressed: 

1, In line 172-174, LN229 cells have IC50 of 10.07, 6.20 and 3.81 uM at 24, 48 and 72 hours, respectively. But, the author use “50 and 100 nM concentrations of UM-164 were used to treat glioma cells in subsequent cell experiments.” Why?

2, Is the HA cell line a human microglia cell line? What is the full name of HA?

3, In figure 6B, this tumor growth is slowly. In the control group, the tumors did not appear to grow even after 42 days. Why?

4, The method description is not clear and detailed. In Figure 1C, the vector control group had approximately 300 colonies. But in 4C, the control group had 200 colonies, I think the author seed the same number cells, but why are there different results?

5, The maker isn’t show in the original WB figures.

Reviewer 2 Report

Dear authors:

This is a well-written manuscript with sufficient scientific evidence to prove the conclusion/title of the paper. There are a few of unclear/uncertain data sets that will need to be revised.

1. I would suggest a repeat of cell cycle on the SF539 cell line, if you do not see the S phase change very well, you can certainly increase the concentration. I also suggest you to fix the cells by paraformaldehyde (3.7% in water) for 15 minutes, which is better than Ethanol fixation in my experience so that you do not see too much necrosis population of the cell cycle detection.

2. It is unclear why the figure 3B showed heat map in reverse way for the repeated group. Could you explain in the discussion/data section?

3. It is true that p38 over-expression recover the expression of AXL and CYR61 more significantly than the transfection of src. However, this can be more affected by many factors including the transfection population, cell's ability to tolerate expression of certain proteins, and many more. So it may not necessarily mean p38 is more weighed than src in UM-164 treatment. I would suggest to change the argument to UM-164 inhibitor hits both src and p38 which could affect the YAP oncogenic activity in glioma. 

4. I suggest adding another paragraph of YAP related to src or p38 in the introduction. It would be more comprehensive to the paper if you could explain how YAP translocation to cytoplasm from nucleus would prevent its activity. For example, YAP could bind with TEAD in the nucleus for transcription of AXL and CYR61, and YAP in the cytoplasm could bind with TAZ for proteasomal degradation.

Reviewer 3 Report

c-Src has been shown to play a pivotal role in breast cancer progression, metastasis, and angiogenesis. UM-164, a novel c-Src inhibitor, resulted in an anti–triple-negative breast cancer role through specifically binding inactive conformation of its target kinases. This paper reports the molecular mechanism by which UM-164 inhibits p38 MAPK pathway, which induces the inhibition of glioma cells proliferation, migration and cell cycle arrest. The authors show that UM-164 inhibits p38 kinase activity by reduction of phosphorylation levels. They also refer to the mechanism by an inhibitory effect on YAP activation and ectopic expression of wide type YAP can rescue UM-164-induced anti-proliferative effect. This is an interesting paper, but there are some issues that should be addressed.

1.     In the introduction part, authors should introduce Hippo-YAP pathway and the relationship with p38 MAPK pathway in cancer;

2.     In figure 1c, how do the authors calculate the colony number? It looks like the number in the left pane is not consistent with quantification data. The number of LN229-0 should be more than 2 folds, compare with SF539-0;

3.     In figure 1e, UM-164 looks like affect the size of cells, not the number;

4.     In figure 3b, do authors find some p38-targeting genes among the down-regulated list;

5.     In figure 3e, the resolution of image is not high enough to make a convincing case.

Round 2

Reviewer 1 Report

Accept in present form

Author Response

Thank you for your suggestions and consideration on this manuscript.